# The Agricultural Potential of a Region with Semi-Dry, Warm and Temperate Subhumid Climate Diversity through Agroecological Zoning

Edgar Vladimir Gutiérrez Castorena [1], Gustavo Andrés Ramírez Gómez [2,*] and Carlos Alberto Ortíz Solorio [2]

1   Facultad de Agronomía, Universidad Autónoma de Nuevo León, Francisco I. Madero S/n, Ex hacienda El Canadá, General Escobedo CP 66050, Nuevo León, Mexico
2   Colegio de Postgraduados, Campus Montecillos. Car. Mexico-Texcoco km 36.5, Montecillos CP 56230, Edo. Mexico, Mexico
*   Correspondence: gustavormz440@gmail.com; Tel.: +52-8116647827

**Abstract:** The sustainability of the natural resources used in agricultural production is essential to meet the future food needs of the population. It is necessary to understand the characteristics of climate and soil changes through agroclimatic zoning models, even with non-existent or limited climatic and edaphic databases, to avoid a decline in production. The objective of the study was to determine the accuracy of the Global Agroecological Zoning (GAEZ), ECOCROP and Papadakis models for major cereals, vegetables and fruit trees in the state of Nuevo León, Mexico, using the databases of climatic stations and soil profiles collected by INEGI with random sampling in the field. The model with the best projection was ECOCROP, which predicted 37,609 km$^2$ of irrigated area for sorghum and 34,796 km$^2$ for wheat, in addition to identifying by soil characteristics rainfed areas with higher suitability for beans measuring 8470 km$^2$ and orange measuring 6175 km$^2$ with zoning predictions based on field information. In conclusion, the thematic maps obtained with ECOCROP had an accuracy greater than 50% for more than half of the crops analyzed, making it the best method for the study area. Therefore, the food production decisions of the producers must be directed towards cereal crops based on the projected area; however, it is necessary to establish an updating program and generate edaphoclimatic databases, updating thematic soil and climate maps with models that support the projections verified in the field.

**Keywords:** agroecological zones; northeastern Mexico; agroclimatic suitability; spline

## 1. Introduction

Climate change will impact agricultural production, according to predictions of food availability during the decades of the 21st century [1]. Furthermore, there will be an increase in pests and diseases [2], imbalances in the hydrological cycle in agriculture [3], a reduction in surface and groundwater bodies [4] and increased agricultural land use change due to industrialization [5]. Consequently, the location and identification of areas with agricultural productive potential benefit both the state and the producer by reducing yield gaps of local or regional importance [6]. The zoning of agricultural areas is one of the main tools for protecting and managing the intensive production that safeguards food security [7]. Similarly, there should be a focus on both rainfed and irrigated production systems that provide alternative solutions to practical production problems [8].

Initially, zoning agroecology (AEZ) focused on crops' location and productive capacity in rainfed systems worldwide based on edaphoclimatic information generated according to efficiency areas by experts in a classification system, delimiting their geospatial suitability [9–12]. The principle was to identify the climatic factors (temperature, precipitation, environmental humidity, among others) that influence crop development by zoning homogeneous agroclimatic regions with similar climatic characteristics and to evaluate

productive capacity as a function of soil quality, water availability and natural resources. AEZ in Asia (Bangladesh, Malaysia, Philippines, Thailand, Indonesia, China, Sri Lanka and Vietnam) had great potential to detect changes in agricultural productivity under different climate scenarios [13]. Meanwhile, the first studies in Mexico were made for maize, bean and sorghum crops as a function of climate and altitude. These were integrated into soil units with slope variables and textural classes in edaphic inventories [14]. However, the lack of continuous thematic maps hindered the recording and analysis of agronomic and climatic variables in relation to their temporal behavior space [15] and crop diversification in agricultural areas.

The latest version of AEZ, called global agroecological zoning (GAEZ), integrates the spatial projection of yield potential in irrigation and rainfed areas by thematic mapping into a geographic information system (GIS) and remote sensing [16–18], with the interplay between soil characteristics, climate, relief and land use and management being used to identify and cartograph areas with productive potential or constraints and to suggest crops and management practices that are appropriate to those areas. For this purpose, using climate data records collected over more than 30 years [19–21] allows the flexibility of the information applied to large territorial extensions or areas with scarce information. However, these advantages achieved have limitations for large-scale thematic maps [22] with a resolution of 400 km$^2$ (20 km × 20 km) [12] because the topographies of rugged landscapes are more challenging to predict due to the variability and the low density of stations [20], which limits the data collection to a minimum of information at the regional level [23].

On the other hand, Papadakis proposed another method based on a matrix model of precipitation and temperature variables to estimate adaptive ranges through climatic classification with projection to suitability and management [24]. This method offers advantages over traditional climatic classification systems by including and considering the local variations in climate and soil conditions. It also provides a more detailed and nuanced understanding of the factors that determine crop performance, allowing for more targeted and effective management strategies, with some information on the projection of areas to diverse climate scenarios [25]. In addition, thematic soil maps recommend areas suitable for irrigated crops [26], highlighting the ease of using moisture indices to delineate areas suitable for irrigated or rainfed agriculture [27,28]. Likewise, the method has greater accessibility in the application and interpretation of thematic maps, which facilitates decision making in government agencies, the private sector and rural communities [29], and a low requirement for climatological data [30] and the geographic distribution of pathogens [31].

ECOCROP is a global database with information on cultivated plant species around the world and their tolerance to environmental factors developed to analyze the data of gene banks and find genetic, ecological and geographical patterns and distribution patterns. It is easy to use without the need for extensive knowledge of the use of geographic information systems [32] while the climate maps are generated from information collected from weather stations from 1970 to 2000 with a resolution of 1 km$^2$ [33,34]; despite this, the model produces reliable results in the simulation of crops at a global and regional level due to the flexibility of the parameterization of the climatic requirements of the crops in the database, which can be adapted to conditions that limit zoning [35]. In Mexico, the use of ECOCROP is essential, used in the location and distribution of ecological niches of wild plants [36] and in relation to the resilience of ecosystems [37] and delimitation of crops such as peanuts, potatoes and capuí [38]; sunflower, wine and rosemary [39]; or legumes and pseudo-cereals [40]. Consequently, the uncertainty is associated with the precision of the field data [41], standardization of the parametric variables in the analysis, modeling and comparison of the patterns of extrapolation in regions of the same country [22], the limitations in the biophysical parameters of the crops and underestimation of the suitability for long-cycle crops [42] due to the lack of maps that show the suitability by soil characteristics in relation to weather events such as droughts or floods [43]. The few

climatic stations, the lack of complete data in more than 70% of these in the state and the lack of thematic maps with edaphic properties increase the need for zoning that provides certainty about the productive agricultural potential. Therefore, the objective of this study is to evaluate the accuracy of the GAEZ, Papadakis and ECOCROP methodologies, as well as the use of soil maps and field sampling, in determining the zoning of crops with the potential for production in the state of Nuevo León, Mexico.

## 2. Materials and Methods

### 2.1. Study Area and Geospatial Data

The state of Nuevo León is located between the geographic coordinates 98°25′18″ and 101°12′24″ west longitude and 23°09′46″ and 27°47′57″ north latitude, covering an area of 64,156.2 km$^2$ in 51 municipal entities [44]. The minimum and maximum temperatures vary significantly with the change of season due to the area being located within three physiographic regions, the Great Plains of North America (GPNA), the Coastal Plain of the Northern Gulf (CPNG) and the Sierra Madre Oriental (SMO) [45], which generate subtropical regions and high-pressure systems of the Pacific and the Atlantic [46] (Figure 1).

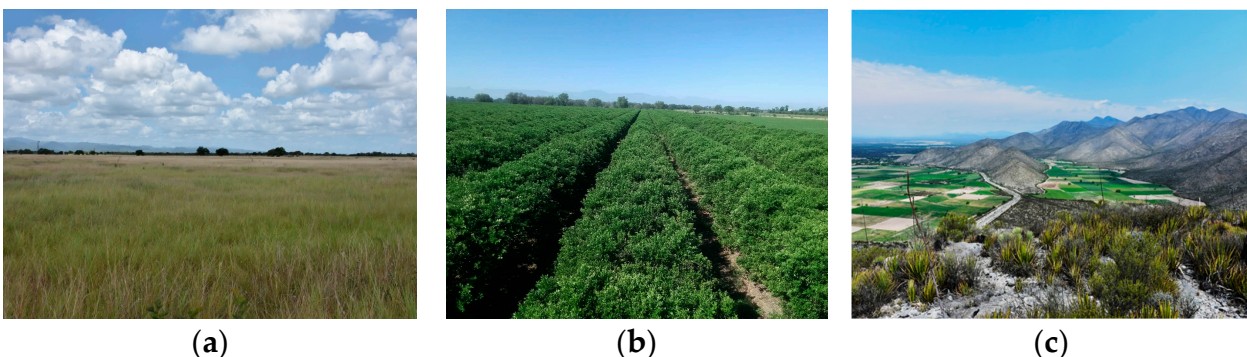

| (**a**) | (**b**) | (**c**) |

**Figure 1.** Study area: (**a**) grasslands on the Great Plains of North America (GPNA), (**b**) orange trees on the Northern Gulf Coastal Plain (CPNG) and (**c**) other crops on the Eastern Sierra Madre (SMO).

The physiographic province of the Great Plains of North America begins in the north of the state of Nuevo León and covers 19 municipalities; the natural vegetation consists of prairies, steppes and grasslands, with alternating plains and hills with a smooth relief composed of sedimentary rocks. It has a wide variety of climates throughout the year, with frigid winters and sweltering and humid summers with prolonged periods of drought; the maximum and minimum temperature ranges between 29.4 °C and 16.8 °C, and the annual average temperature is 22.7 °C. The average annual rainfall in the region is 520 mm. The soils with more cover are categorized according to their type; some are Calcisol, Luvisol, Leptosol, Vertisol, Chernozem and Solonchak [47].

On the other hand, the Northern Gulf Coastal Plain presents features of an emerged coast, with coastal lagoons interrupted by isolated mountain ranges and hills with agricultural and livestock activity covering the lower lands with extensive alluvial plains in 27 municipalities in the subprovinces of Plains and Lomerios. The reported maximum and minimum temperatures vary between 29.5 °C and 15.8 °C, and there is an annual average temperature of 22.6 °C. The average annual precipitation is 757 mm. The soil types with the greatest cover are Phaeozem, Vertisol Chernozem, Regosol and Calcisol.

Meanwhile, the Sierra Madre Oriental comprises a narrow and elongated mountain range 1350 km long with an amplitude between 80 and 100 km and an altitude between 2000 and 3000 m. The temperature ranges between 26.3 °C and 10.5 °C, with an average annual temperature of 18.4 °C. The average annual precipitation is 486 mm. The soil types with the greatest coverage are Calcisol, Chernozem, Cambisol, Leptosol and Solonchak. The sedimentary rock is of marine origin and belongs to two physiographic subprovinces

in the south of the state, the Gran Sierra Plegada, where 16 municipalities are located, and the Sierras y Llanuras Occidentales, with coverage of about five municipalities.

The crops selected for edaphoclimatic zoning had to meet the following criteria: be cultivated in more than 95% of the existing agricultural area in the entity; have availability of government information (beans, grain corn, orange, potato, sorghum and wheat of grain); and have a discrepancy between the database and null production of the crop in the field. Each crop's edaphoclimatic requirements were considered in the optimal condition reported by the ECOCROP database [48].

The criteria to establish crop suitability were based on the stipulation by the agroecological zone approach that determines land potential [11] and were modified into three categories: Very Suitable (>80% of the maximum), Suitable (from 80% to 40%) and Marginally Suitable (from 40% to 0%); these percentages were obtained from the suitability of the climate and soil in relation to the crops' yields. Additionally, the meteorological stations reported by the National Meteorological Service (SMN) [49] were selected based on the information reported in their databases between 1981 and 2010. These stations were used to prepare the thematic maps, with the following requirements: they had to (1) have data from a minimum period of 20 years; (2) be currently in operation; (3) have at least 70% complete data.

### 2.2. Interpolation of Climatic Variables and Soil Properties

The soil samples were collected to a depth of 30 cm and analyzed with the analytical procedures established in [50]. On the other hand, the interviews with the producers addressed questions relating to aspects such as the maximum and the minimum crop yields in the cycle or agricultural year, planting dates, harvest dates, frost occurrence and precipitation patterns and were used to create a field verification database.

The interpolation of the precipitation data was carried out using ordinary kriging [51], while, in terms of temperature, the inverse distance weighting was applied due to the high significance level of the results in earlier studies [52,53].

Currently, Mexico does not generate national thematic maps, or lacks them in some regions, to reveal the chemical and physical properties of soils with the precision and accuracy of soil unit cartography, which impedes the evaluation of the resource for agricultural purposes. Consequently, a geostatistical spline method was used to interpolate edaphic soil properties such as pH, electrical conductivity (EC), textural class and internal drainage with the representation of distribution patterns over large surface areas [54,55]. The database information was recorded for the soil profiles of 114 sampling points [47] sampled to a depth of 30 cm by aqueous extract.

### 2.3. Description of Agroclimatic Zoning

The agroclimatic methodologies of GAEZ, ECOCROP and Papadakis were employed to delimit the crops' productive potential in irrigated and rainfed soils. On the one hand, the precipitation and temperature variable data were employed to generate thematic maps of the rainfed crop (ECOCROP-S, GAEZ-S); in contrast, the values of the temperature variable were analyzed only in the modality of irrigation (ECOCROP-R and GAEZ-R) since its value is necessary to satisfy the water requirements of crops. On the other hand, the Papadakis methodology, which considers the recommendations for management, was employed to determine the suitability of the crops in rainfed areas (PAPADAKIS-S) or under irrigation (PAPADAKIS-R), establishing the need for physical spaces and hydraulic infrastructure as prerequisites for the production system.

### 2.4. Agroecological Zoning (Modified GAEZ)

The original GAEZ model consists of seven modules describing the formulas to obtain each variable with their information requirements [16]. However, in the present study, only modules one to five were partially used since the necessary data to apply the procedure in its original form were unavailable (Figure 2).

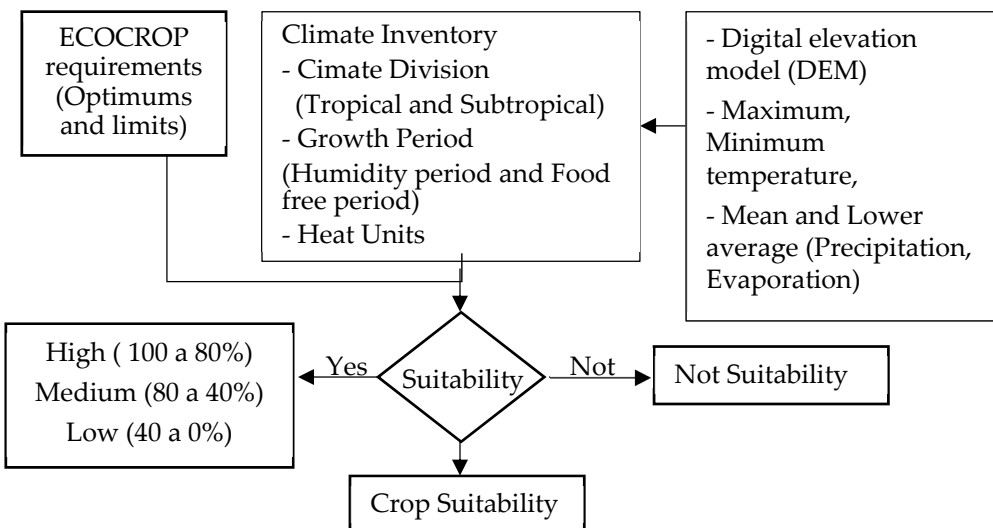

**Figure 2.** Suitability procedure by the modified GAEZ method.

*2.5. Zoning with the ECOCROP Method*

The ECOCROP method (Figure 3) was used to determine the suitability of the proposed crops for the state by evaluating their productivity potential based on temperature and precipitation data for the consecutive months of the crop cycle [56].

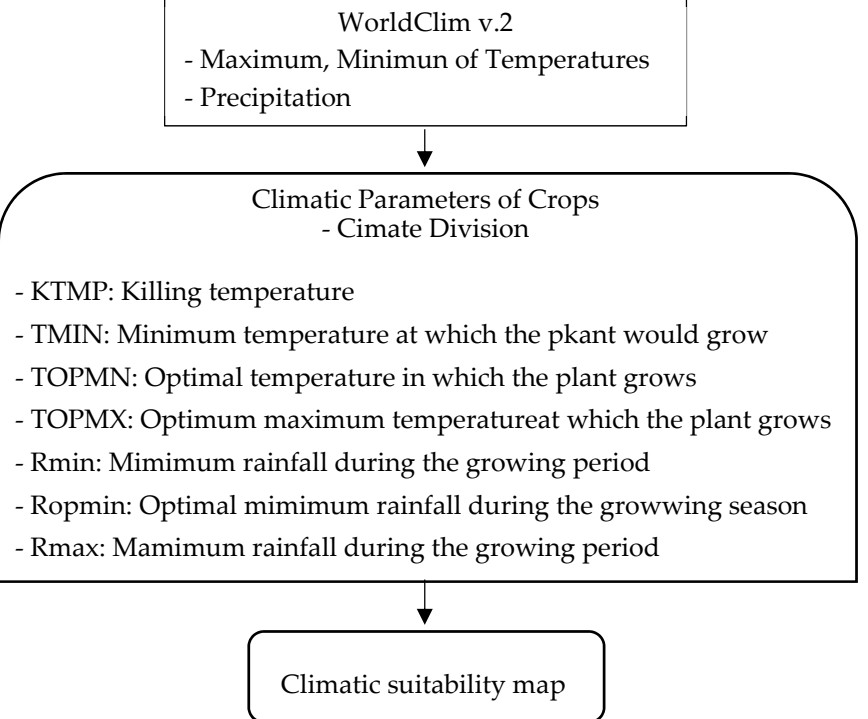

**Figure 3.** The procedure of the ECOCROP method.

*2.6. Agroclimatic Zoning Using the Papadakis Method*

The thematic maps of climatic classification generated by the Papadakis methodology are based on the type of summer, winter and water regime, proposing climatic divisions and subdivisions, which have, in turn, assignment to a codification of up to two decimal places associated with recommendations that determine the suitability of the crops [57] (Figure 4).

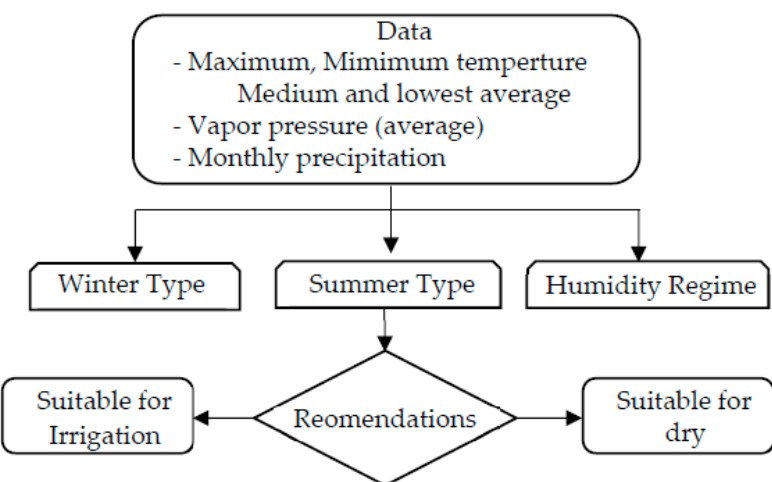

**Figure 4.** Papadakis zoning procedure method.

*2.7. The Edaphoclimatic Suitability of Crops*

The thematic zoning maps prepared with the three methodologies were finally superimposed with the suitability of the edaphic variables for each crop (Figure 5) directly checked in the field in the 11 municipalities of Allende, Anáhuac, Aramberri, Bustamante, Cadereyta Jiménez, Dr. Arroyo, Galeana, General Terán, Lampazos del Naranjo, Linares and Montemorelos.

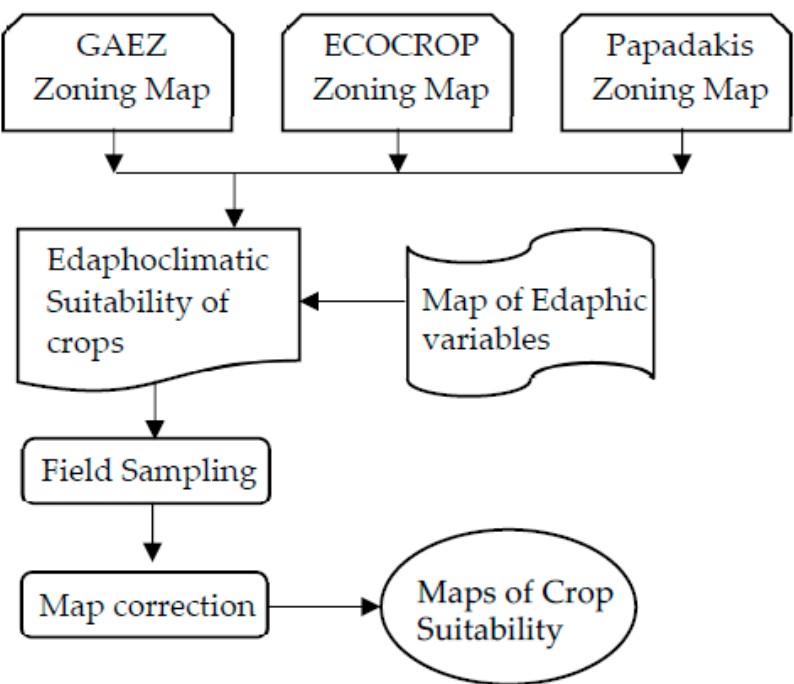

**Figure 5.** Procedure for the edaphoclimatic suitability of crops.

## 3. Results

*3.1. Weather Stations, Sampling Sites and Crops*

Based on the selection criteria of the meteorological stations, out of a total of 103, 42% were discarded, and only 60 databases were taken into account, of which eight were located outside the geopolitical limited, and 51 were distributed throughout the federal entity; 11 stations were located in the North American Plains, 30 in the Sierra Madre Oriental and 19 in the Northern Gulf Coastal Plain. On the other hand, 75 sites of surface horizons were analyzed and compared with the pedological profile reported by INEGI. In addition,

48 consultations were made with producers of different socio-economic levels, focusing on crop diversification (Figure 6).

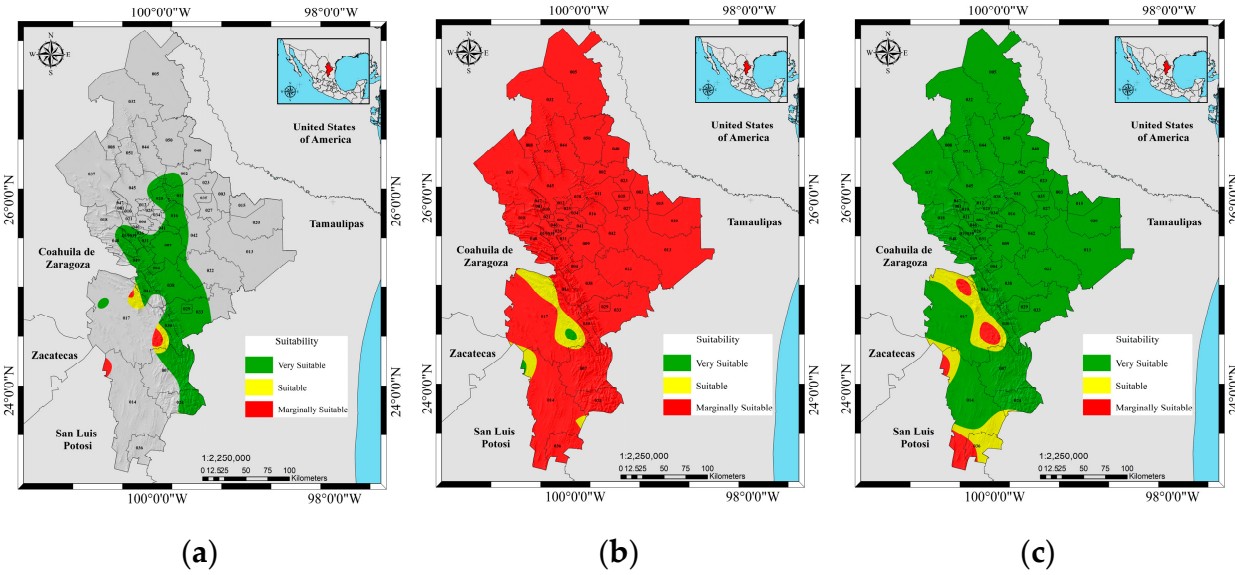

**Figure 6.** Spatial data: (**a**) keys to municipalities and weather stations, (**b**) crop sampling and physiographic provinces and (**c**) soil sampling points.

### 3.2. Zoning Using the GAEZ Method

The projection of agricultural land for rainfed crops was 20 to 21% of the state's area, indicating a marginal aptitude class; this was due to the methodology, which excludes areas with limited productivity from the beginning (Figure 7a). However, irrigated sorghum production was reclassified and assigned to medium and highly suitable classes in the three physiographic regions (GPNA, CPNG and SMO) (Figure 7c). Meanwhile, for orange cultivation on irrigated land, a state agricultural area of 91% was projected for the GPNA and CPNG regions (north and center of the state) (Figure 7b). However, maize and other crops were registered in 4% and 5% of the area with a medium and marginal suitability class, respectively.

**Figure 7.** Agroclimatic suitability for the cultivation of (**a**) orange (GAEZ-S), (**b**) maize (GAEZ-R) and (**c**) orange (GAEZ-R).

### 3.3. Zoning Using the ECOCROP Method

Based on climatic information reported by WorldClim2.1 [33], the agroclimatic zoning obtained by the methodology proposed by ECOCROP projected an agricultural area of 35% state coverage for the cultivation of rainfed sorghum with an adequate suitability class; however, it decreased for rainfed corn (Figure 8a), with a high aptitude class located mainly in the GPNA and CPNG regions (northeast and east of the state). On the other hand, in irrigated lands, corn cultivation extended 30% from the northeast to the southeast of the state, with a very suitable class, and corn was considered by producers as the crop with the second highest yield under these conditions, assigning the crops of corn (Figure 8b), potato (Figure 8c) and wheat (Figure 8d) as the main crops that occupied an agricultural area greater than 75% with a very suitable class for the three physiographic regions of the state.

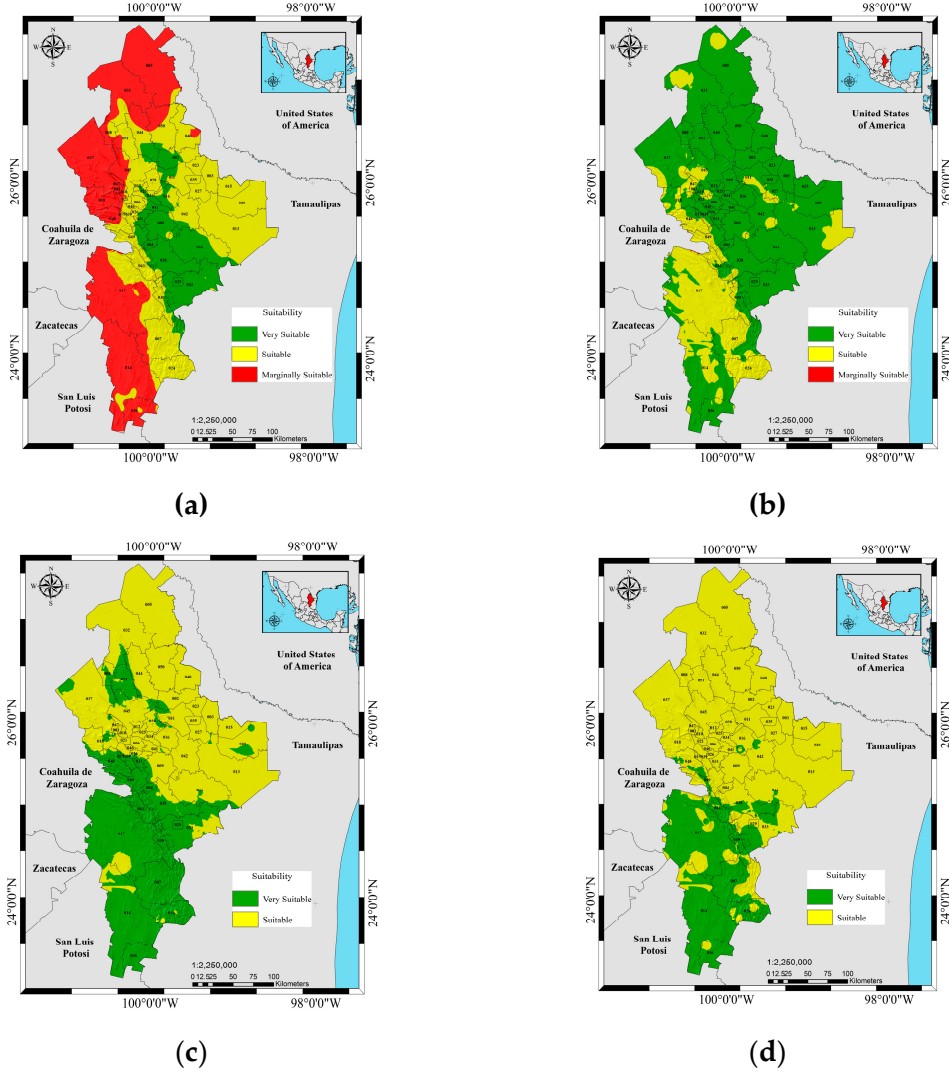

**Figure 8.** Agroclimatic suitability in rainfed or irrigation land crops: (**a**) corn cultivation in rainfed land, (**b**) corn cultivation on irrigation land, (**c**) potato crop on irrigation land and (**d**) wheat crop on irrigation land.

### 3.4. Zoning Using the Papadakis Method

The climatic thematic maps generated with the Papadakis method indicated that the cotton zone (G) predominated in the summer type classification, with 65% of the area in the state's entire northern and central zone (Figure 9a). Meanwhile, 66% of the area located from the north to the south corresponded to the citrus zone (Ci), typically in winter, as a

monsoonal type of water regime (Mo, mo) (Figure 9c) prevails in 57% of the state located in the GPNA regions to the north and SMO to the south.

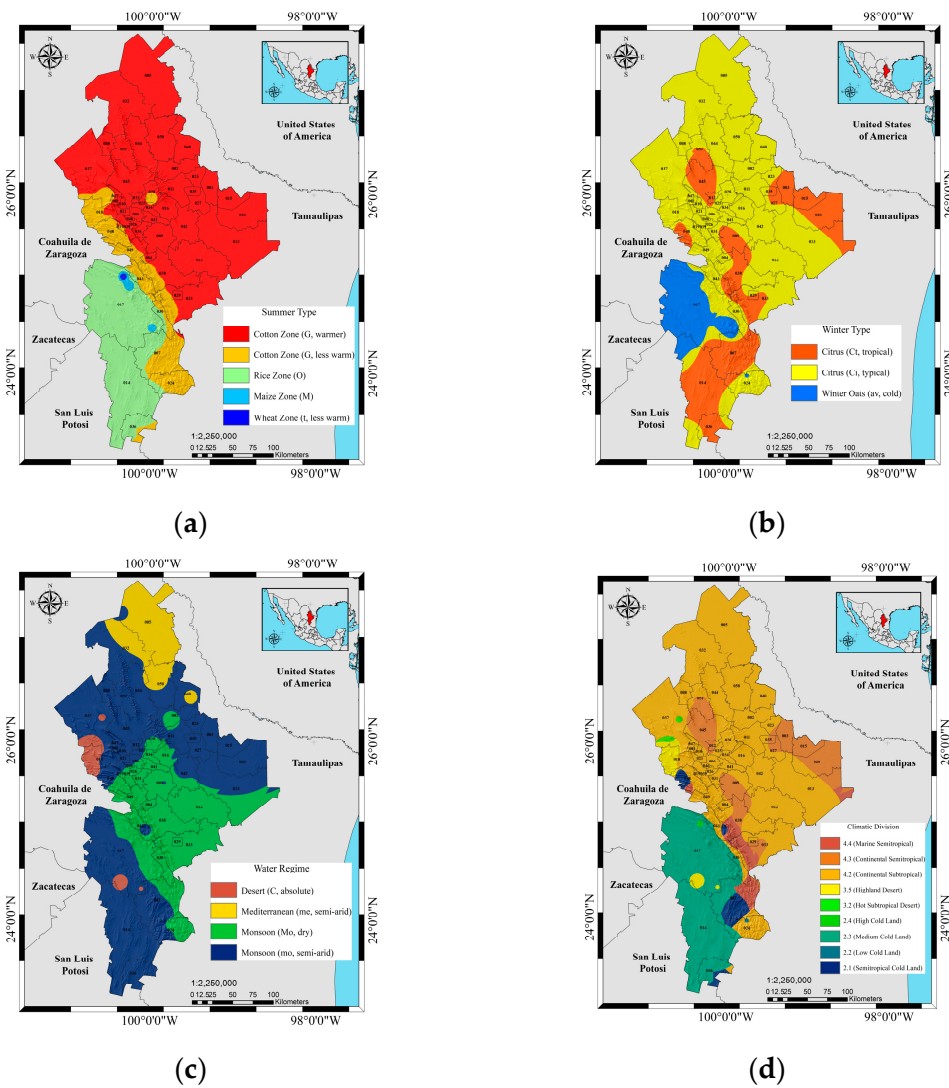

**Figure 9.** Climatic thematic maps of the state of Nuevo León: (**a**) summer type, (**b**) winter type, (**c**) water regime and (**d**) climatic division.

On the other hand, the climatic division of the state (Figure 9d) was made up of a total of seventeen subgroups belonging to three main groups: Cold Land (2), Desert (3) and Subtropical (4). The first climatic group had the largest area (58% of the state) due to the GPNA physiographic region reporting a highland desert (group 3.5, subdivisions: 3.53 and 3.54) located to the northeast, with a small area of 2% to the south in the SMO region; these are regions where irrigated crops such as corn and sorghum were reported to have a high suitability class in most agricultural lands. Other sites corresponded to the continental Subtropical subdivision (group: 4.2; climatic subdivisions: 4.21, 4.22, 4.23 and 4.24) located in the physiographic region of the SMO, where both rainfed sorghum and irrigated oranges had a high aptitude class. Finally, the medium Cold Land group (group: 2.3; subdivisions: 2.34, 2.35, 2.36 and 2.38), located in the south of the state (with an area of 22%) in the SMO physiographic region, was where both rainfed corn (area of 6%) and irrigated land corn, potato and wheat had a surface of 21% and a high class of suitability (Figure 10). Since the Papadakis methodology did not recommend cultivating beans, it was decided that beans would be excluded from the suitability evaluation.

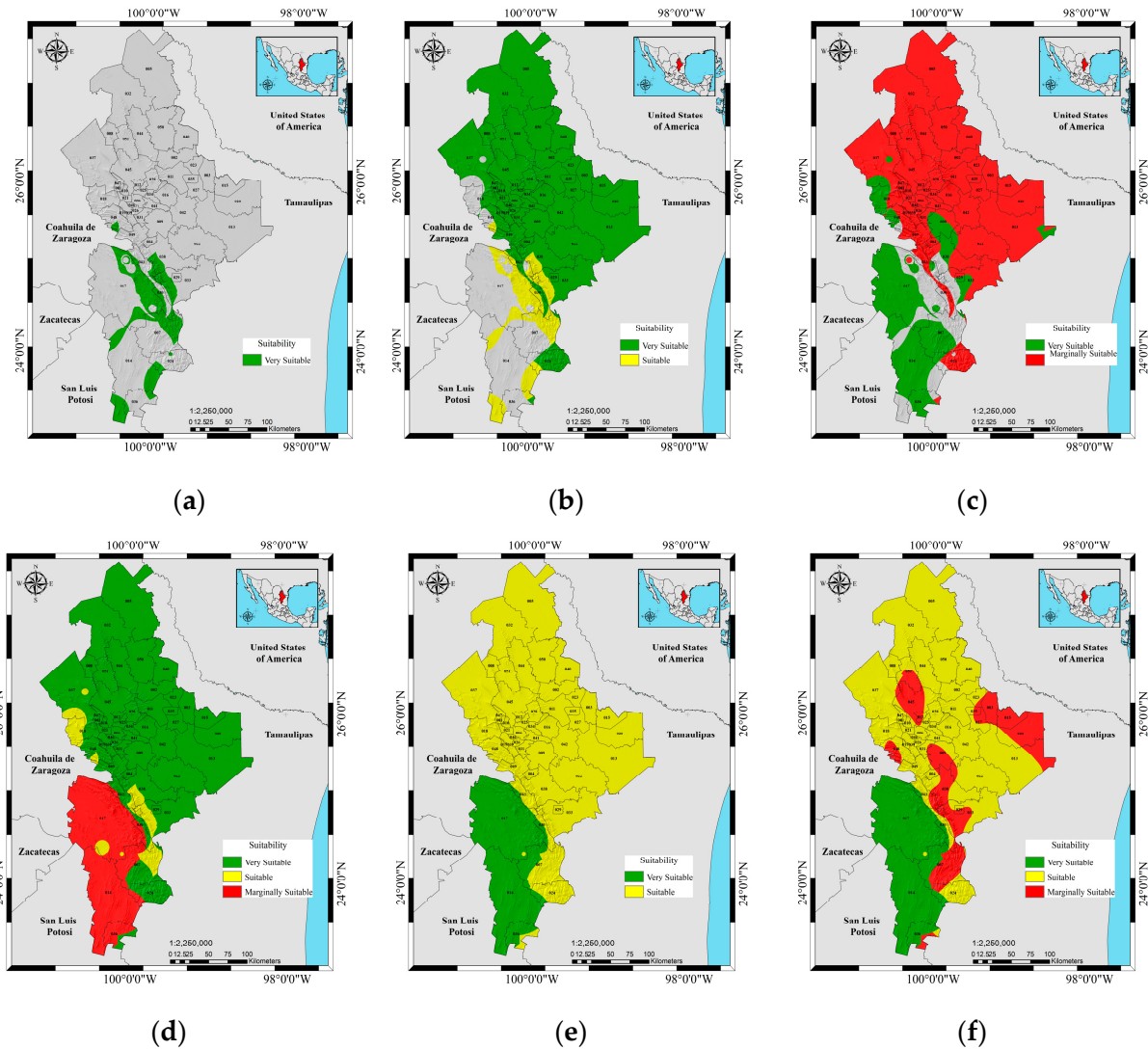

**Figure 10.** Suitability of Papadakis climatic groups and subdivisions for dryland or irrigation land. (**a**) Rainfed corn, (**b**) rainfed sorghum, (**c**) orange on irrigation land, (**d**) corn on irrigation land, (**e**) potato on irrigation land and (**f**) wheat on irrigation land.

### 3.5. Edaphic Suitability

Since soil is considered another limiting factor for agricultural production, since it can increase or reduce the yield per unit area, the INEGI database was used to produce thematic maps of the spatial distribution of soil types, showing that 80% of the agricultural area located in the GPNA and SMO physiographic areas was alkaline or slightly alkaline, with a pH between 8 and 9, and another 20%, located in the CPNG area, was neutral, with a pH of 7 (Figure 11a). In comparison, 89% of the agricultural area was non-saline (0 to 1 dS m$^{-1}$) and was located in the GPNA and CPNG regions (Figure 11b). Furthermore, the texture of the agricultural soils was most commonly loamy in 62% of the area (Figure 11c) and poorly to moderately drained in 52% (Figure 11d).

The thematic maps of the edaphic suitability of the crops of interest to the state showed that the physiographic region of the North American Great Plains was very suitable, and the Sierra Madre Oriental was suitable for six crops. It was reported that the sorghum crop had excellent soil suitability in 72% of the area of the state (Figure 12e), while bean and potato crops had excellent soil suitability in 45% of the area (Figure 12a,d) in the three physiographic regions. On the other hand, maize, orange and wheat were scattered over 30% of the surface of all the municipalities (Figure 12a−c). Therefore, the suitability of all

crops was divided into two classes, Very and Moderately Suitable, for more than 50% of the surface, with beans, wheat and potatoes having a more excellent suitability, and maize, orange and sorghum covering a larger area with medium suitability.

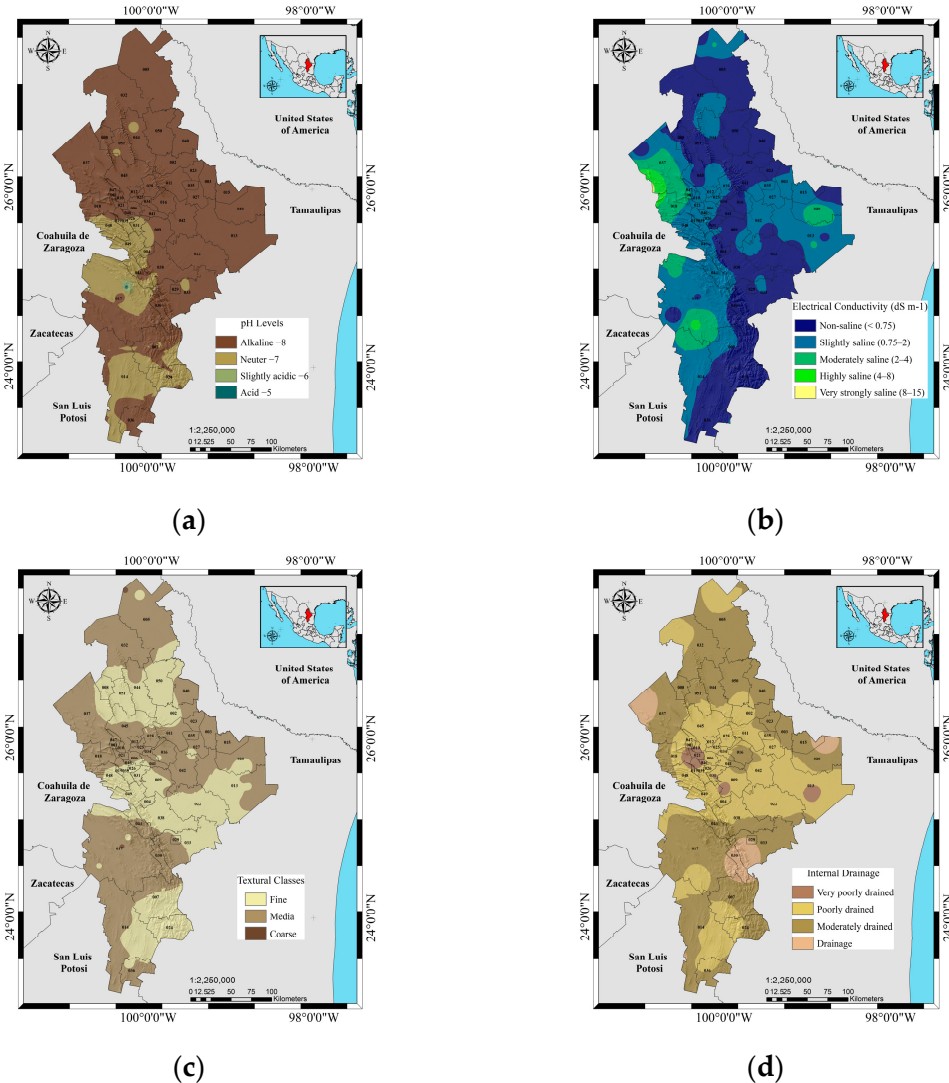

**Figure 11.** Soil properties in Nuevo León: (**a**) soil pH, (**b**) soil electrical conductivity, (**c**) soil texture and (**d**) internal soil drainage.

*3.6. Field Check*

Grower interviews provided historical yield information for 23 crops, including avocado, oats, peanut, lemon, forage maize, moringa, star and praetorian grass, cabbage, forage sorghum and red tomato and indicated that the most common crops were: maize in 35 sampling sites, wheat in 25 sites, sorghum in 17, beans in 12, oranges in 11 and potatoes in 4.

When comparing the information in the database with information in the field, these maps showed frost-free periods (FFP) and rainy periods for rainfed crops with the highest precision for FFP onset (48%); however, there was no precision more significant than 7% for the end of the FFP. In contrast, for the beginning of the rainy period, 0% precision was obtained and, for the end, 32% precision. In addition, it is essential to note that there was a similarity in the yields of some crops, such as sorghum and wheat, in different areas and production systems (Table 1) for both modalities.

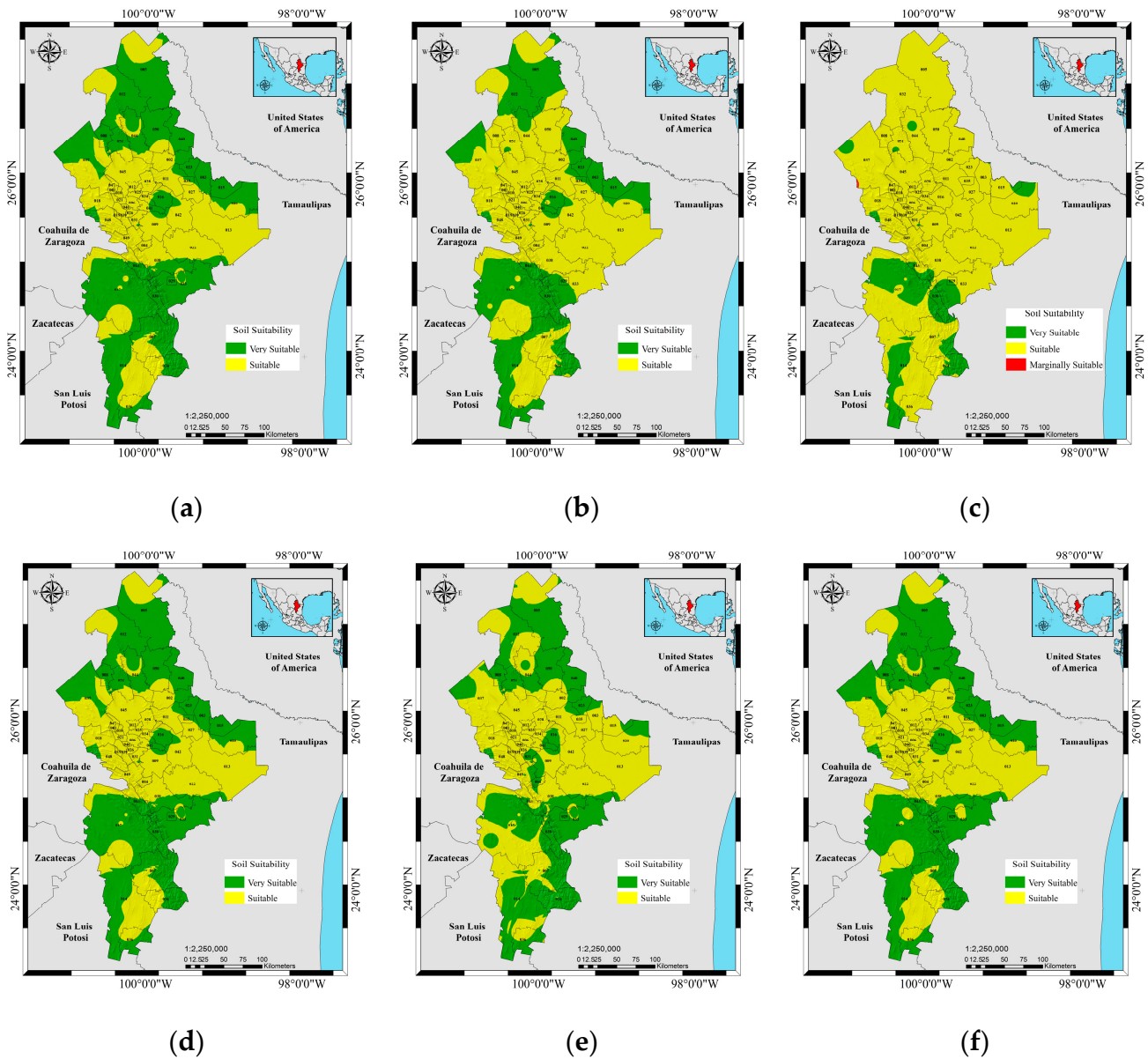

**Figure 12.** Edaphic suitability of crops: (**a**) bean, (**b**) corn, (**c**) orange, (**d**) potato, (**e**) sorghum and (**f**) wheat.

**Table 1.** Crop yield based on suitability.

| Crop | By Municipality (Code) | | Dryland Suitability (t ha $^{-1}$) | | | Irrigation Suitability (t ha $^{-1}$) | | |
|---|---|---|---|---|---|---|---|---|
| | Dryland | Irrigation | VS | S | Ma | VS | S | Ma |
| Bean | 038 | 032 | 2–1.6 | 1.6–0.6 | 0.6–0.1 | 3–2.4 | 2.4–1 | 1–0.2 |
| Corn | 022 | 017 | 3–2.4 | 2.4–1 | 1–0.2 | 11–8.8 | 8.8–3.5 | 3.5–0.7 |
| Orange | 038 | 038 | 12–9.6 | 9.6–3.8 | 3.8–0.8 | 45–36 | 36–14.4 | 14.4–2.9 |
| Potato | N/A | 017 | | N/A | | 47–37.6 | 37.6–15 | 15–3 |
| Sorghum | 022 | 009 | 4–3.2 | 3.2–1.3 | 1.3–0.3 | 4–3.2 | 3.2–1.3 | 1.3–0.3 |
| Wheat | 022 | 017 | 5–4 | 4–1.6 | 1.6–0.3 | 5–4 | 4–1.6 | 1.6–0.3 |

VS = Very Suitable; S = Suitable; Ma = Marginally Suitable; N/A = not applicable. Code of municipalities: 009 = Cadereyta Jiménez, 017 = Galeana, 022 = Gral. Terán, 032 = Lampazos de Naranjo and 038 = Montemorelos.

The agroclimatic thematic maps with higher accuracy were those produced by the ECOCROP methodology for rainfed crops, which showed an accuracy of 50% for orange cultivation in 31% of the agricultural area. At the same time, the thematic maps for agricultural systems in irrigated areas showed that the accuracies for orange, sorghum and wheat crops were greater than 55, 53 and 68%, respectively, for 90% of the state's cultivated area. On the other hand, the GAEZ methodology estimated a precision of 58% only for dryland bean cultivation, which corresponded to 71% of the cultivated area (Table 2).

**Table 2.** Agroclimatic precision for different crops.

| Crop | Irrigation | | | Dryland | | |
|---|---|---|---|---|---|---|
| | ECOCROP-R | GAEZ-R | PAPADAKIS | ECOCROP-S | GAEZ-S | PAPADAKIS |
| Bean | 8 | 33 | N/A | 25 | 58 | N/A |
| Corn | 11 | 17 | 20 | 29 | 26 | 31 |
| Orange | 55 | 18 | 27 | 55 | 18 | 0 |
| Potato | 0 | 0 | 0 | 0 | 0 | 0 |
| Sorghum | 53 | 41 | 6 | 47 | 35 | 41 |
| Wheat | 68 | 28 | 24 | 12 | 8 | 0 |

N/A = not applicable.

The edaphic thematic map registered a precision in the agronomic variables of texture and internal drainage of 64% and 23%, respectively, and proposed a coefficient of determination for soil chemical variables of 0.02 for pH and 0.38 for electrical conductivity. Thus, it was decided to use the variables with a precision of 50% and a coefficient of determination of at least 0.45, which characterized the soils for 62% of the agricultural area located in the physiographic region of CPNG and SMO (Figure 13a) as Very Suitable for sorghum cultivation, which meant that 100% of the agricultural area was very suitable (Figure 13b).

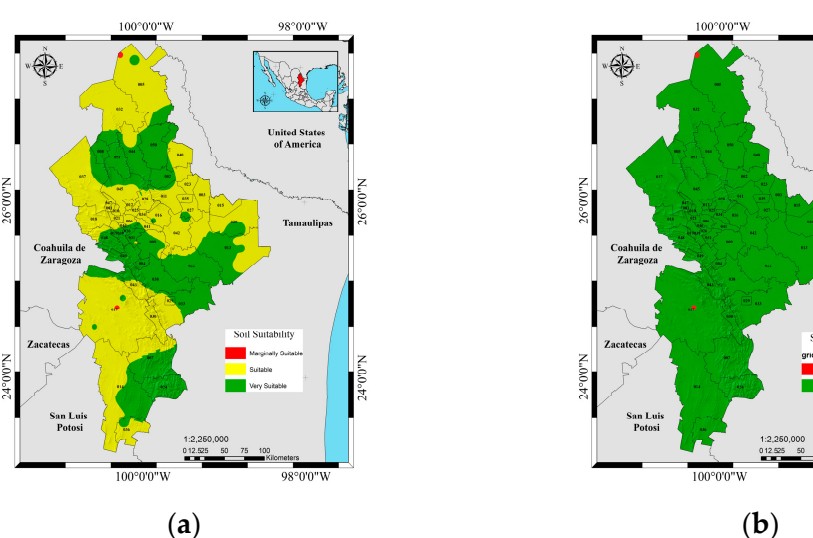

(**a**)                                  (**b**)

**Figure 13.** Edaphic suitability of the corrected crops: (**a**) for beans, maize, orange, potato and wheat and (**b**) sorghum.

### 3.7. The Edaphoclimatic Suitability of Crops

Thematic characterization maps for edaphoclimatic suitability of crops showed that the methods with the highest precision obtained in the rainfed modality were the ECOCROP-S for orange and wheat crops, estimating a precision of 55% and 64%, respectively, in the three physiographic regions, and the GAEZ-S model for wheat, with a precision of 56%. Thematic maps demonstrating the highest precision were obtained using ECOCROP-R for orange cultivation, with a precision of 55%, and GAEZ-R and PAPADAKIS-R for wheat cultivation, with a precision of 64%. (Figure 14). The relationship between suitability and yield increased the precision of the thematic maps for orange and wheat crops, establishing them as an alternative to diversify agricultural lands in a rainfed and irrigated production

systems by projecting them onto the three physiographic regions (Figure 14), as indicated by the ECOCROP and GAEZ methodologies. However, the wheat crop had a more remarkable aptitude for production in the Sierra Madre Oriental region (Figure 14e).

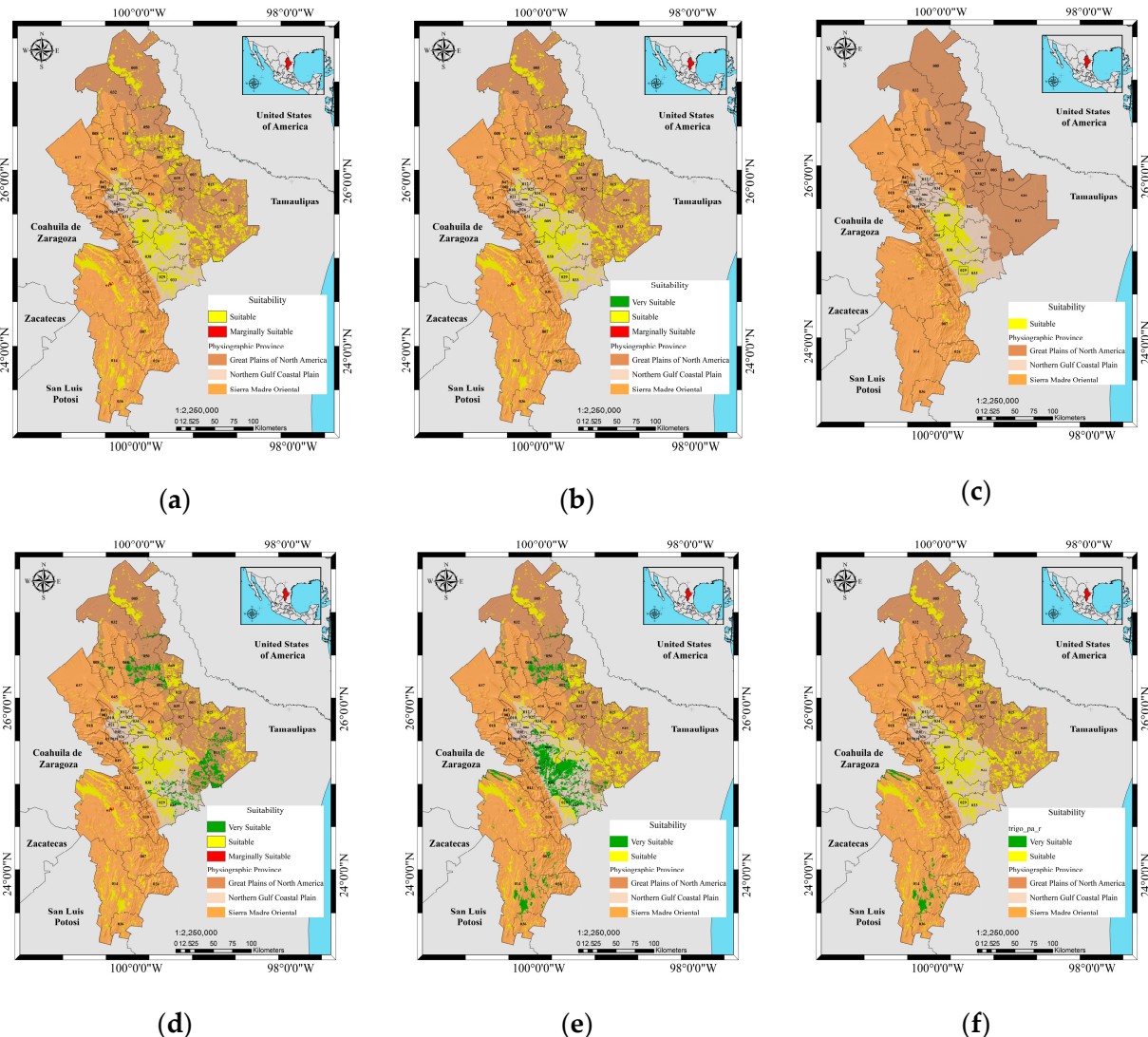

**Figure 14.** Edaphoclimatic suitability of the methodologies with greater precision in lands for agricultural use: (**a**) orange crop (ECOCROP-S), (**b**) wheat (GAEZ-S), (**c**) wheat (ECOCROP-R), (**d**) orange (ECOCROP-R), (**e**) wheat (ECOCROP-R) and (**f**) wheat (PAPADAKIS-R).

## 4. Discussion

The zoning established by GAEZ had certain drawbacks due to the compatibility of the information required to generate the thematic maps proposed by the original methodology's seven modules [58]. This had implications in the climatic analysis by thermal and hydric regime, which was necessary for analysis corresponding to climate and soil and was motivated by the geomorphology of the landscape of the three physiographic regions of the state. This was exemplified in areas with folds, plains and mountains at very short distances, making it difficult to establish climatic stations with optimal spatial distribution. Other implications were related to the estimation of soil evapotranspiration, which depends on factors that were difficult to estimate for the entire study area, leading to the use of models that limited the use of those parameters.

Although the second module determined the potential yield and yield with eco-physiological limits [16], it was suggested that we implement processes according to the physiography of the state, for example, direct interviews with producers in each region

determined the maximum real yields obtained per unit of surface. The perspective and interpretation of the information collected should be consistent with reality for both a rainfed and irrigated production system in terms of the maximum yields of each crop determined, indicating that the edaphoclimatic factors are close to the potential yield as the estimated yield approaches the estimated maximum. However, it could only be partially solved, since the yield problem was still being estimated with information generated from climate and soil according to the suitability of the place.

Continuing with the implications in zoning, it was found that the factors affecting the crop were implicit in the yield [59], determining the potential reduction in yield. Therefore, emphasis was placed on considering edaphic properties based on the drawbacks of assigning them to a fitness class. However, the information generated uncertainty due to the lack of corroboration between the database and the field information, specifically in the omission of data on the specific characteristics of the horizons in the pedological profiles and the focus only on the arable layer. In this way, the main difficulty of the methodology was to obtain each of the necessary variables for both climate and soil, which was satisfactorily resolved by generating accuracies greater than 50%. It is hoped that future studies will have bases of more precise data, according to the reality.

On the other hand, the zoning registered with the Papadakis methodology required a greater variety of agroclimatic parameters (climate classifications, heat units, development day, evapotranspiration, among others) [30] and the reporting of specific categories for each crop in terms of climatic variables. The zoning was similar to that reported by ECOCROP. However, the establishment of strategies such as soil suitability ranking was considered to minimize difficulties in estimating specific data for variables such as vapor pressure values, thus avoiding models outside the established procedure that require climatic data.

In the present investigation, climatology was limited to the subdivision of climatic groups with specific requirements, avoiding discrepancies between them. For example, in the specific case of the subdivision of crops according to water requirements (rainfed or irrigated), various recommendations were made specifically for areas where the classification was correctly applied. The methodology was not modified when working in areas with or without irrigation.

Agroecological zoning is feasible but limited in Nuevo León due to the low coverage of weather stations. Therefore, although the suitability of the crops coincided with a production focused on basic grains [60], the potential yield differed when compared with the information provided by producers, showing an important gap between the possible performance of suitability rating and reality.

Consequently, it is necessary to use methodologies designed based on the requirements of the crops that best fit the available information. It is also necessary to increase the precision of agroclimatic studies to generate new geospatial details and the suitability of the same crops by reducing the lack of climatic data. In this case, the ECOCROP method meets these requirements due to its low number of climatic variables (precipitation and temperature) and its having the largest number of maps for each crop in rainfed and irrigated modalities [61]. The ECOCROP method has a precision >50%. However, it has limitations caused by the scale of observation (1 km$^2$) and limitations of the information of its database collected until the year 2000 on mountainous geomorphology, such as that of the Sierra Madre Oriental, and thematic maps by suitability for cultivation. Therefore, it was more unwieldy to determine the suitability of crops, in addition to having low precision when comparing the field information with the maps of Nuevo León. In addition, soil profile maps are not recommended because they had a maximum accuracy of 55% and a maximum R$^2$ of 0.48. Therefore, it is recommended to continue with the implementation of new models to generate more detailed information on growth and frost-free periods with field studies at the municipal level for the proposed crops.

## 5. Conclusions

The most appropriate methodology for the agroecological zoning of the state of Nuevo León was ECOCROP due to the precision obtained in the thematic suitability maps and the projected performance in zones with limited climatic databases. Thus, the production of basic cereals should be considered in those projected locations with greater suitability in the three physiographic regions.

The physiographic regions studied differ drastically in terms of landscape geomorphology and climate, which are the main factors that determine the suitability of crops in areas with very specific climates. For this reason, it is necessary to establish observation and testing sites that provide information on both climate and soil. In this way, the behavior of the crops can be predicted with greater certainty, and thematic maps can be drawn with greater precision.

**Author Contributions:** Conceptualization, data curation, formal analysis, G.A.R.G. and C.A.O.S.; methodology, resources, funding acquisition, supervision, E.V.G.C. and C.A.O.S.; visualization, writing—original draft, writing—review and editing, E.V.G.C., G.A.R.G. and C.A.O.S. All authors have read and agreed to the published version of the manuscript.

**Funding:** This research received no external funding.

**Institutional Review Board Statement:** Not applicable.

**Informed Consent Statement:** Not applicable.

**Data Availability Statement:** The data presented in this study are available on request from the corresponding author.

**Acknowledgments:** The National Council of Science and Technology (CONACYT) for the financing granted through the scholarship and the Produce Nuevo León A.C. foundation and the SADER delegation through the Rural Development Districts and their CADERs for help with logistics and contact with the producers.

**Conflicts of Interest:** The authors declare no conflict of interest.

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
