# Peer review of "The Agricultural Potential of a Region with Semi-Dry, Warm and Temperate Subhumid Climate Diversity through Agroecological Zoning"

_sustainability, doi:10.3390/su15129491_

Round 1
Reviewer 1 Report
General review:
Overall, the publication provides a highly intriguing subject, and the specific research appears to have some important findings for the field's research community. The suggested paper is exceptionally well-written and makes excellent use of the English language. However, the paper’s spelling and grammar sides is very poor which need to improved. To fix these little errors, the authors should review the work again.
You find in below the Points which has been found in this manuscript, where the authors need to revision it:
1. Line 92: (priority crops for the state of Nuevo León, Mexico.2. Materials and Methods) …. please remove this: 2. Materials and Methods.
2. Line 94: 2.1. Geospatial data: the authors should be adding here the map which represent the study area and the samples.
3. Line 131 to 133: for geostatistical techniques: what’s technique used in this study.
4. Line 133 to 135: please rephrasing this sentence: the pH and EC for water and not for soil.
5. Figure 1,2,3: is blurry
6. Figure 5,6,7,8,9,10,11,12,13: the authors should be redrawn this figures and tike same tails of the text. The coordinates are blurry. Also, the authors need to but here the originally maps (With this resolution, the Paper maybe reject)
7. Table 1:
7.1. in the title part: separate between the first line and the second with a line. (ex: between the municipality (Code) and dryland Irrigation
7.2. please replace the = by : .
8. Table 2: take just the first capital letter.
9. Discussion: missing
10. Conclusion: does not exist
Respecting the above mention requirements, the manuscript in current version does not meet the publication criteria. I would suggest significant improvement is necessary and the paper may be again submitted to the journal.
Author Response
Major changes were made due to reviewers' comments, thank you very much for your feedback.

Reviewer 2 Report
Dear Editor,
Thank you for inviting me as a reviewer for manuscript “Agricultural potential of a region with semi-dry, warm and temperate subhumid climate diversity through agroecological zoning”. The article is presents an interesting topic. The authors provided good interpretations. However, hoping to assist the authors in their research efforts, I provide several suggestions for improving the presented work.
Major comments
- It is better to include more effective keywords to attract readers.
- The paper lacks to describe related works as well as a state of the art of research including the methods used in literature making not clear the originality and soundness of the research work.
- Authors should enhance the introduction section by adding more recent and relevant articles related. The authors need to discuss their contributions compared to those in related papers. From that discussion author will provide a proper research question and the contribution of the paper. Contribution of the paper should be clearly stated.
- The methodology section lacks the laws and fundamentals for the used model.
- According to the distribution of the used data, the use of Kriging interpolation method is not suitable. I recommend using the spline interpolation method.
- The authors did not provide sufficient details on the data used. More information should be provided about the sources, versions, accuracy and the limitations of use these data.
- In the section Discussion, do include the relevance of your work and explain why it is important, i.e., its relevance and added value. Compare the results presented in the manuscript with results presented in relevant papers published earlier by other authors. Please cite those papers properly and add a proper reference.
- Conclusions needs to be written to highlight the novelty and/or findings of this study.
Best regards
Author Response

(The authors gave the same response as above.)

Reviewer 3 Report
Please see attached

Author Response

(The authors gave the same response as above.)

Reviewer 4 Report
My major concerns are:
1. The objective of the paper is not well described in the introduction and the authors didnt explain the novelty of their work.
2. Methodology needs a major improvement. There is a lack of information on the methods you applied. You should explain each method in detail.
3. The data and their sources are not clear. Please modify it.
4. The discussion doe not include information regarding advantages and limitations of the research compared to other studies.
5. Please include the conclusion section.
Attached you can find my detailed comments on the manuscript

Author Response

(The authors gave the same response as above.)

Round 2
Reviewer 1 Report
After the reviewing of this manuscript, I appreciate the response of the authors to my comments as I suggested. With this current version, I can consider this manuscript to published it in this journal.
Reviewer 2 Report
Dear Editor,
Greeting
Thank you again for inviting me as a reviewer for this interesting paper. The authors have responded well to my concerns, and I think the manuscript has a sufficient contribution for publication. Accordingly, I recommend acceptance of the manuscript.
Congratulations